# Continual Learning with Filter Atom Swapping

**Zichen Miao, Ze Wang, Wei Chen, Qiang Qiu**
Department of ECE
Purdue University
`{miaoz, wang5026, chen2732, qqiu}@purdue.edu`

## Abstract

Continual learning has been widely studied in recent years to resolve the *catastrophic forgetting* of deep neural networks. In this paper, we first enforce a low-rank filter subspace by decomposing convolutional filters within each network layer over a small set of filter atoms. Then, we perform continual learning with filter atom swapping. In other words, we learn for each task a new filter subspace for each convolutional layer, i.e., hundreds of parameters as filter atoms, but keep subspace coefficients shared across tasks. By maintaining a small footprint memory of filter atoms, we can easily archive models for past tasks to avoid forgetting. The effectiveness of this simple scheme for continual learning is illustrated both empirically and theoretically. The proposed atom swapping framework further enables flexible and efficient model ensemble with members selected within task or across tasks to improve the performance in different continual learning settings. Being validated on multiple benchmark datasets with different convolutional network structures, the proposed method outperforms the state-of-the-art methods in both accuracy and scalability.

## 1 Introduction

Humans keep acquiring new concepts without forgetting crucial ones in the past. To endow intelligent agents with the same ability of long-term knowledge accumulation, continual learning (CL) has been intensively studied in recent years. In continual learning, an agent learns from a sequence of tasks, with the goal of gaining knowledge of each new task while preserving the capacity for resolving the old ones, therefore to avoid *catastrophic forgetting*. The recent advances of CL mainly follow several directions. One popular category among them is to maintain an external memory of the original images (Robins, 1995; Rebuffi et al., 2017), synthesized images (Shin et al., 2017), or parameter gradients (Lopez-Paz & Ranzato, 2017) for archiving the past. These memory-based methods often suffer from heavy memory footprints, while still forgetting about the previous tasks to some extent.

Motivated by the literature on subspace modeling of tasks (Evgeniou & Pontil, 2007; Maurer et al., 2013; Zhang & Yang, 2021; Romera-Paredes et al., 2013; Kumar & Daume III, 2012), in this paper, we propose to learn for each task a new filter subspace for each convolutional layer, i.e., hundreds of parameters as filter atoms, but keep subspace coefficients shared across tasks. In other words, in a CNN, we enforce a low-rank filter subspace by decomposing convolutional filters within each network layer over a small set of filter atoms. Then, we perform continual learning by simply swapping filter atoms for each task. The effectiveness of our approach is empirically validated and further explained theoretically with an excess risk bound analysis.

With the proposed approach, we can faithfully remember the past by only maintaining an atom memory with small footprint to archive task-specific filter atoms. Any previously learned CNN models can now be exactly recovered by multiplying the task-shared coefficients with the task-specific atoms, which can be retrieved efficiently from the atom memory. Thus, the introduced filter atom memory permits historical knowledge to be recalled with a guarantee against forgetting. Comparing with state-of-the-art memory-based CL methods, our approach requires storing for each task only some tiny size filter atoms, which in total are typically much smaller than the size of exemplars in memory-based methods (Rebuffi et al., 2017; Prabhu et al., 2020), and therefore potentially supports continual learning on a large scale.

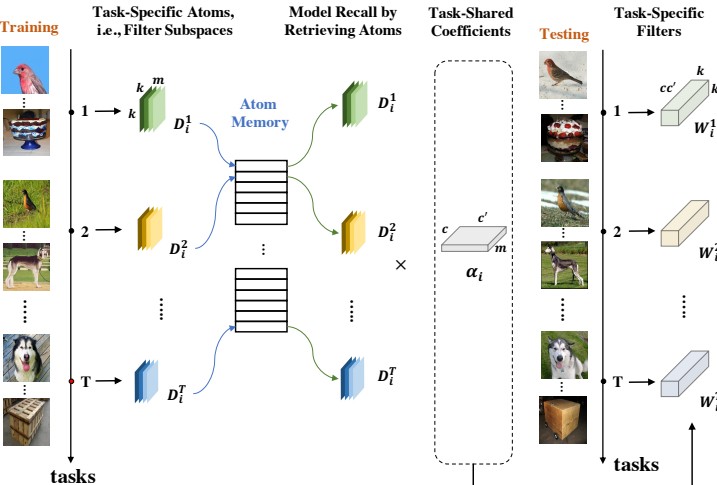

Figure 1: Illustration of the proposed continual learning method with filter atom swapping. Within each CNN layer, we decompose a filter $\mathbf{W}_i \in \mathbb{R}^{c \times c' \times k \times k}$ over a filter subspace spanned by $m$ filter atoms $\mathbf{D}_i \in \mathbb{R}^{m \times k \times k}$ as $\mathbf{W}_i = \boldsymbol{\alpha}_i \mathbf{D}_i$, where $\boldsymbol{\alpha}_i \in \mathbb{R}^{c \times c' \times m}$ are the subspace coefficients, $c$ and $c'$ are the number of input and output channels, $k$ is the spatial size of each atom. With task-shared coefficients, we learn for each task a new filter subspace as filter atoms, and store those atoms, typically a few hundred of parameters, in a small footprint atom memory. At time $T$, we can recall the past model at $t$ ($t < T$) through filter reconstruction $\mathbf{W}_i^t = \boldsymbol{\alpha}_i \mathbf{D}_i^t$, with $\mathbf{D}_i^t$ fetched from the atom memory, to fully recover the previous model.

Our atom swapping continual learning framework can effectively support both inter-task and intra-task model ensemble to further enhance performance in different continual learning settings: First, *inter-task ensemble* utilizes the relevant past knowledge to boost the present task performance. To ensure that ensemble with past models can affect the current task positively, we only select a relevant subset of past models. With a life-long learning scenario in mind, we choose to assess task relevancy simply based on the filter subspace distance, which can be on-the-fly computed here via the Grassmann distance (Absil et al., 2004) among task-specific filter subspaces. Second, *intra-task ensemble* can be adopted in class-incremental setting to help task prediction with the minimal-entropy criterion. Usually, ensemble members are instantiated as independent CNNs, and their learning and inference are conducted separately for dissimilarity. However, this will lead to significant increase in training and inference time and memory usage. We address this problem by creating within a task multiple virtual members in a single CNN model by simply maintaining several groups of filter atoms in each layer. In this way, different intra-task members are integrated into a single network, while learning and inference can be conducted efficiently with group convolution.

We validate our simple yet effective approach on several continual learning benchmarks such as MNIST, CIFAR100, and miniImageNet under both class-incremental and task-incremental settings, and observe competitive results against state-of-the-art methods on all benchmarks with far less memory usage.

We summarize our contributions as follows,

- We learn for each task a new filter subspace for each layer, and keep subspace coefficients shared across tasks.
- We maintain a small footprint filter atom memory that can faithfully archive past knowledge with a guarantee against forgetting in a highly scalable way.
- We adopt an *inter-task ensemble* for the present task by recalling past models based on an on-the-fly calculated task relevancy under the task-incremental setting.
- We propose an *intra-task ensemble* for the class-incremental setting by creating multiple virtual members in a single CNN model through different groups of filter atoms per layer.

## 2 MOTIVATION

We are motivated by the literature on task subspace modeling (Evgeniou & Pontil, 2007; Maurer et al., 2013; Zhang & Yang, 2021; Romera-Paredes et al., 2013; Kumar & Daume III, 2012), where it is commonly assumed that task parameters lie in a low dimensional subspace, so that tasks can be modeled as a set of latent basis tasks and their linear combinations. The latent basis tasks and

the respective linear combinations are often obtained via alternative optimization by fixing one and optimizing the other (Kumar & Daume III, 2012).

In our continual learning setting, we model tasks using convolutional neural networks (CNNs). Following (Qiu et al., 2018), we decompose a convolutional filter $\mathbf{W}_i \in \mathbb{R}^{c \times c' \times k \times k}$ for the $i$-th layer over $m$ filter atoms $\mathbf{D}_i \in \mathbb{R}^{m \times k \times k}$, linearly combined by coefficient $\boldsymbol{\alpha}_i \in \mathbb{R}^{c \times c' \times m}$, where $c$ and $c'$ are the number of input and output channels, $k$ is the spatial size of each atom. This can be written as $\mathbf{W}_i = \boldsymbol{\alpha}_i \times \mathbf{D}_i$. Note that this decomposition distributes a filter $\mathbf{W}_i$ into two imbalanced parts: $\boldsymbol{\alpha}_i$ for channel mixing with $mcc'$ parameters, and light-weight $\mathbf{D}_i$ for spatial filtering with only $mk^2$ entries. In all, we use $\mathbf{W} = \boldsymbol{\alpha} \times \mathbf{D}$ to denote the filters decomposition in a model with $l$ convolutional layers, where $\mathbf{W} = \{\mathbf{W}_i\}_{i=1}^l, \boldsymbol{\alpha} = \{\boldsymbol{\alpha}_i\}_{i=1}^l, \mathbf{D} = \{\mathbf{D}_i\}_{i=1}^l$ indicate all filters, coefficients, and atoms respectively.

It is easy to observe that, within each CNN layer, we can borrow from the task subspace modeling methodology by creating a set of latent basis tasks through filter atoms $\mathbf{D}_i$ and their linear combinations as atom coefficients $\boldsymbol{\alpha}_i$. The linear combination coefficients $\boldsymbol{\alpha}_i$ are learned on the first task jointly with the first group of atoms, and then atoms for subsequent tasks are alternatively optimized by fixing $\boldsymbol{\alpha}_i$.

As illustrated in the subsequent sections, this seemingly under-fitting method not only enables an efficient way to faithfully archive past models with a guarantee against forgetting, but also supports efficient inter-task and intra-task model ensemble to further improve the performance.

## 3 METHODOLOGY

We consider the problem of learning $T$ tasks sequentially. Formally, we denote the data distribution that associates with the $t$-th task as $\mathcal{D}^t = (\mathcal{X}^t, \mathcal{Y}^t)$, $t \in \{1, 2, ..., T\}$, from which a dataset $D^t = \{\mathbf{x}_t^i, \mathbf{y}_t^i\}_{i=1}^{N_t}$ is sampled for training. The goal of continual learning is to minimize the statistical risk of all seen tasks given no access to data $D^t$ from previous tasks $t \leq T$ (Delange et al., 2021):

$$\sum_{t=1}^{T} \mathbb{E}_{(\mathcal{X}^t, \mathcal{Y}^t)}[\mathcal{L}(\mathcal{F}^t(\mathcal{X}^t; \theta), \mathcal{Y}^t)], \tag{1}$$

where $\mathcal{L}$ denotes the risk function, $\mathcal{F}^t(\cdot; \theta)$ is the model for task $t$ with parameter $\theta$. Continual learning with a guarantee against forgetting can be achieved by storing learned parameters entirely in an external memory $\mathcal{M} = \{\theta^t\}_{t=1}^T$ given $\mathcal{F}^t(\cdot; \theta) = \mathcal{F}(\cdot; \theta^t)$, thus any previous model can be completely recovered by retrieving the corresponding parameters from the memory. However, such a straightforward solution based on parameter memory suffers severely on its poor scalability due to the large size of modern deep neural networks and the potentially long task sequence. We address the scalability issue and achieve guaranteed non-forgetting by decomposing convolutional filters in a CNN into task-specific filter atoms and task-shared coefficients. Then only hundreds of parameters per task need to be stored in an atom memory to guarantee non-forgetting. This proposed approach allows efficient inter-task and intra-task ensemble to further boost performance.

### 3.1 A SCALABLE APPROACH AGAINST FORGETTING

In CNNs, catastrophic forgetting occurs when a model learned from a sequence of past tasks is updated in favor of the current task, resulting in significant performance degradation. A straightforward solution, as mentioned before, is to archive $\mathbf{W}^t$ in an external memory $\mathcal{M} = \{\mathbf{W}^t\}_{t=1}^T$, and the representation space of any previous model can be faithfully recalled by memory retrieval. However, as deep CNNs contain great amount of parameters in $\mathbf{W}^t$, this simple solution scales poorly with the number of tasks $T$. On the other hand, storing part of parameters, or small subsets of data for parameter flashback cannot avoid forgetting completely (Sarwar et al., 2019; Yoon et al., 2018).

This dilemma can be resolved with the proposed filter atom decomposition, as shown in Fig. 1. With the filter decomposition described in Sec. 2, by storing task-specific filter atoms $\mathbf{D}^t$ into memory, and enforcing a task-shared coefficients $\boldsymbol{\alpha}$, the model archives the entire knowledge for each time point. We refer to the memory for storing atoms as the *atom memory*, $\mathcal{M}_{\mathbf{D}} = \{\mathbf{D}^t\}_{t=1}^T, \mathbf{D}^t \in \mathbb{R}^{m \times k \times k}$, with each $\mathbf{D}^t$ learned in the $t$-th task with empirical risk minimization,

$$\arg\min_{\mathbf{D}^t} \sum_{i=1}^{N_t} \mathcal{L}(\mathcal{F}(\mathbf{x}_i^t; \boldsymbol{\alpha}, \mathbf{D}^t), \mathbf{y}_i^t). \tag{2}$$

The task-shared coefficients $\boldsymbol{\alpha}$ are learned on the first task jointly with the first group of atoms,

$$\underset{\mathbf{D}^1, \boldsymbol{\alpha}}{\arg\min} \sum_{i=1}^{N_1} \mathcal{L}(\mathcal{F}(\mathbf{x}_i^1; \boldsymbol{\alpha}, \mathbf{D}^1), \mathbf{y}_i^1). \tag{3}$$

In this way, we can guarantee that the statistical risk for a previous task $t$ at any time point remains the same as we can recall *faithfully* the past model by multiplying the stored atoms with the task-shared coefficients,

$$\mathbb{E}_{(\mathcal{X}^t, \mathcal{Y}^t)}[\mathcal{L}(\mathcal{F}^t(\mathcal{X}^t; \theta), \mathcal{Y}^t)] = \mathbb{E}_{(\mathcal{X}^t, \mathcal{Y}^t)}[\mathcal{L}(\mathcal{F}(\mathcal{X}^t; \boldsymbol{\alpha} \times \mathbf{D}^t), \mathcal{Y}^t)]. \tag{4}$$

**Atom memory scalability.** The proposed atom memory stores a group of atoms per task, which is scalable with increasing number of tasks. Formally, consider a $l$-layer CNN model with associated filters $\mathbf{W} = \{\mathbf{W}_i\}_{i=1}^l$. As mentioned in Sec. 2, each filter can be decomposed as $\mathbf{W}_i = \boldsymbol{\alpha}_i \times \mathbf{D}_i$. The group of atoms for task $t$ can then be denoted as $\mathbf{D}^t = \{\mathbf{D}_i^t\}_{i=1}^l$, which requires a size of $lmk^2$ in storing parameters per task. This typically introduces only a few hundred of parameters for each task to be stored in the atom memory, which potentially supports continual learning on a large scale. Details for scalability comparison are shown in Sec. 5.2.1.

**Analysis of excess risk bound.** With task-shared coefficients $\boldsymbol{\alpha}$, the model for each new task may seemingly expect some degree of underfitting. However, as demonstrated in Section 5, we still observe superior results over the state-of-the-art continual learning methods on all benchmarks we have evaluated. To understand this, we here theoretically analyze the excess risk bound for new tasks in the continual learning setting. For each tasks $t \in \{1, 2, ..., T\}$, model $\mathcal{F}^t(\cdot; \theta)$ consists of representation function $\phi^t$ and prediction function $\mathbf{w}^t$ ($\mathbf{w}^t \in \mathbb{R}^k$). The representation function $\phi^t$ maps an input $\mathbf{x}$ to feature space $Z \subseteq \mathbb{R}^k$. For the analysis purpose, we assume $\phi^t$ is just one convolution layer, which can be decomposed as $\phi^t = \boldsymbol{\alpha} \times \mathbf{D}^t$. Therefore, using the training samples from task $t$, we can solve the following optimization problem from Eq. (2):

$$\min \frac{1}{2N^t} \|\mathbf{y}^t - (\boldsymbol{\alpha} \times \mathbf{D}^t(\mathbf{x}^t))_+ \mathbf{w}^t\|^2 + \frac{\lambda}{2}\|\boldsymbol{\alpha} \times \mathbf{D}^t\|_F^2 + \frac{\lambda}{2}\|\mathbf{w}^t\|_F^2, \tag{5}$$

where we use the mean square error as our loss, $(\cdot)_+$ is ReLU activation $(z)_+ = max\{0, z\}$. The shared coefficients $\boldsymbol{\alpha}$ are learned on the first task jointly with the first group of atoms. Similar to (Du et al., 2020), we also assume that there is a ground-truth optimal representation function $\phi^{t,*}$ and prediction function $\mathbf{w}^{t,*}$ for task $t$.

*Assumption 1* (subgaussian input). There exists $\rho > 0$ such that, for all $t \in \{1, 2, ..., T\}$, the random vector $\bar{x} \sim \bar{p}^t$ is $\rho^2$-subgaussian. The $\bar{p}^t$ is the distribution of samples in task $t$.

*Assumption 2* (oracle network). Assume for task $t \in \{1, 2, ..., T\}$ that $\mathbf{y}^t = (\boldsymbol{\alpha}^* \times \mathbf{D}^{t,*}(\mathbf{x}^t))_+ \mathbf{w}^{t,*} + z^t$ is generated by an oracle network with parameters $\boldsymbol{\alpha}^*$, $\mathbf{D}^{t,*}$, and $\mathbf{w}^{t,*}$. Noise term $z^t \sim \mathcal{N}(0, \sigma^2 I)$.

*Excess risk bound.* We can bound the excess risk of our learned model on the task $t$, *i.e.*, how much our learned model $(\hat{\boldsymbol{\alpha}}, \hat{\mathbf{D}}^t, \hat{\mathbf{w}}^t)$ performs worse than the optimal model $(\boldsymbol{\alpha}^*, \mathbf{D}^{t,*}, \mathbf{w}^{t,*})$ on the task $t$ as follows:

$$ER(\hat{\boldsymbol{\alpha}}, \hat{\mathbf{D}}^t, \hat{\mathbf{w}}^t) = L_{\mathcal{D}^t}(\hat{\boldsymbol{\alpha}}, \hat{\mathbf{D}}^t, \hat{\mathbf{w}}^t) - L_{\mathcal{D}^t}(\boldsymbol{\alpha}^*, \mathbf{D}^{t,*}, \mathbf{w}^{t,*})$$
$$\leq \sigma\bar{R} \cdot \tilde{O}(\frac{\sqrt{Tr(\Sigma)} + \sqrt{\|\Sigma\|_2}}{\sqrt{N^t}}) + \rho^4 \bar{R}^2 \cdot \tilde{O}(\frac{Tr(\Sigma) + \|\Sigma\|_2}{N^t})$$

where $\bar{R} = \frac{1}{2}\|\boldsymbol{\alpha}^* \times \mathbf{D}^{t,*}\|_F^2 + \frac{1}{2}\|\mathbf{w}^{t,*}\|_F^2$, $\Sigma = \mathbb{E}_{\mathbf{x} \sim p}[\mathbf{x}\mathbf{x}^\intercal]$, $N^t$ is the number of training samples. $\tilde{O}$ is the big O notation, and $L_{\mathcal{D}^t}$ is the expected loss with the data distribution $\mathcal{D}^t$. The detailed analysis is provided in the Appendix A.

## 3.2 INTER-TASK MODEL ENSEMBLE

The atom memory not only serves as an efficient way to archive past models with a guarantee against forgetting, but also enables efficient recall of past models for model ensemble to improve the performance in present task. As in Breiman (2001); Lakshminarayanan et al. (2017), ensemble performance increases with the independent level of ensemble members. Motivated by this, model ensemble over time for the current task at time $r$ has a natural advantage that each ensemble member

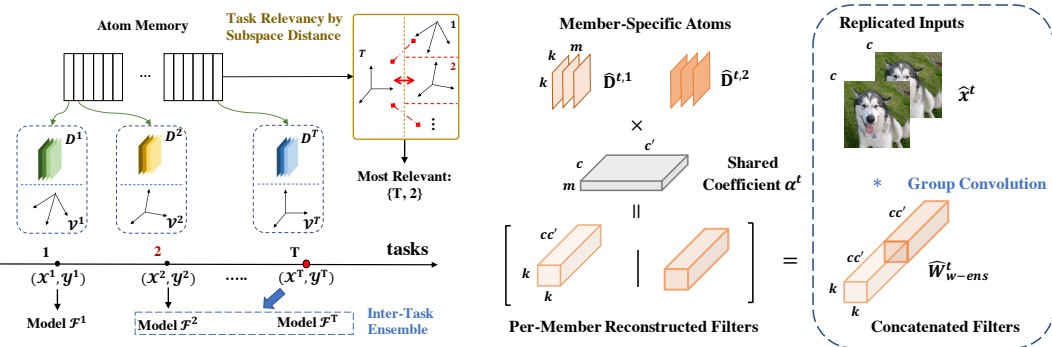

Figure 2: Inter-task ensemble with $E_c = 1$.  Figure 3: Intra-task ensemble with $E_w = 2$.

$\mathcal{F}^t$ ($t \in \{1, 2, ..., r\}$) is learned from a different data distribution $\mathcal{D}^t$. However, most of other methods are not affordable to perform model ensemble across tasks since they often lack an effective and efficient way to recall past models. As illustrated in Fig. 2, with the atom memory, our method can faithfully and rapidly recall past models $\mathcal{F}^t = \mathcal{F}(\cdot; \boldsymbol{\alpha} \times \mathbf{D}^t)$ by simply fetching atoms $\mathbf{D}^t$ from the atom memory; and then perform inter-task model ensemble by constructing a uniformly-weighted mixture model and combine the predictions as (Lakshminarayanan et al., 2017),

$$\mathcal{F}^r_{c-ens}(\mathbf{x}) = \frac{1}{|\mathcal{S}^r| + 1} \sum_{s \in \mathcal{S}^r \cup \{r\}} p_{\mathcal{F}}(\mathbf{y}|\mathbf{x}, \theta^s) = \frac{1}{E_c + 1} \sum_{s \in \mathcal{S}^r \cup \{r\}} \mathcal{F}(\mathbf{x}; \boldsymbol{\alpha} \times \mathbf{D}^s), \qquad (6)$$

where $\mathcal{S}^r$ denotes the index set of previous tasks used for ensemble, and $E_c = |\mathcal{S}^r|$. For classification problem, it corresponds to averaging the predictive probabilities.

In continual learning, not all past models can bring positive effects on the current task. According to (Breiman, 2001; Lakshminarayanan et al., 2017), only the ones that have enough strengths on the current $r$-th task can help enhance the performance as weak learners (Breiman, 2001), which can be selected based on task relevancy. Our assumption is that the more a past task $t$ resembles the current $r$, the better performance will $\mathcal{F}^t$ achieves on the present task. The problem then transforms to evaluating the model similarity effectively and efficiently. Note that to ensure the proposed ensemble method to be scalable across a very long historical task sequence $T$, highly efficient task relevancy assessment is indispensable here.

**Assessing task relevancy by filter subspace distance.**  Task relevancy assessments proposed in (Achille et al., 2019; Zamir et al., 2018) work at the cost of heavy computation, which prevents their efficient applications in continual learning. While measuring the similarities among the learned models can be a straightforward proxy to the measurement of task relevancy, the widely studied methods, e.g., canonical correlation analysis (CCA) (Raghu et al., 2017; Morcos et al., 2018) and centered kernel alignment (CKA) (Kornblith et al., 2019), still introduce considerable computational cost while performing evaluations in the representation space. Directly performing relevancy measurements by calculating the distance of filters works at a highly desirable efficiency, yet can perform poorly without the costly semantic alignments over channels (Raghu et al., 2017). In our approach, thanks to the task-shared coefficients acting as structural regularizations, we show that model similarity measurements can now be efficiently evaluated through direct filter similarity measurements. And as we model filters using task-specific filter subspaces with coefficients shared, model similarity can be further reduced to assessing filter subspace distance via the Grassmann distance (Absil et al., 2004).

Formally, we characterize the filter subspace of the current model $\mathcal{V}^r$ and a past model $\mathcal{V}^t$ as:

$$\mathcal{V}^r = \text{Span}\{\mathcal{B}_1^r, ..., \mathcal{B}_M^r\}, \quad \mathcal{V}^t = \text{Span}\{\mathcal{B}_1^t, ..., \mathcal{B}_M^t\}, \quad \mathcal{V}^r, \mathcal{V}^t \subset \mathbb{R}^L, \qquad (7)$$

where $L = k^2$ is the dimension of kernel space, $\boldsymbol{\mathcal{B}} = \{\mathcal{B}_j\}_{j=1}^M$ are $M$ ($m > M$) linear independent vectors serve as the bases of the filter subspace. We obtain $\boldsymbol{\mathcal{B}}$ by performing singular value decomposition (SVD) to atoms $\mathbf{D} = U\Sigma V^T \in \mathbb{R}^{L \times m}$ and select the first $M$ columns from $U$ that correspond to the top-$M$ singular values. By definition, $\mathcal{V}^r$ and $\mathcal{V}^t$ are points of the Grassmann Manifold (Milnor & Stasheff, 2016), i.e., $\mathcal{V}^r, \mathcal{V}^t \in \text{Grass}(M, L) \triangleq \{M \text{ dimensional subspaces of } \mathbb{R}^L\}$. Then, the Grassmann distance between $\mathcal{V}^r$ and $\mathcal{V}^t$ is defined as,

$$d_M(\mathcal{V}^r, \mathcal{V}^t) = (\sum_j^M \theta_j^2)^{1/2}, \qquad (8)$$

where $\theta_j$ is the $j$-th **principle angle**, which can be calculated by,

$$\theta_j = \arccos(\sigma_j), \text{ with } (\mathcal{B}^r)^T \mathcal{B}^t = U\Sigma V^T, \sigma_j = \Sigma_{jj}. \tag{9}$$

The proposed task relevancy measurement requires merely a SVD to matrices with dimensions lower than $L$. In practice, $k = 3$ so that $L = 9$, indicating that computation of $d_M$ is low. With the proposed efficient task relevancy measurement adopted in the last convolutional layer, we select the most $E_c$ relevant models from previous $r - 1$ ones that support the current task. We provide the correlation analysis of task similarities measured with CCA (Raghu et al., 2017) and the ones assessed by the proposed subspace distance in Appendix. C.5.

Although it is known that the ensemble result increases in the ensemble number $E_c$ (Lakshminarayanan et al., 2017), it does not hold in our setting based on our empirical observation. In fact, determining the $E_c$ is a trade-off between the amount and the relevancy of past knowledge. We thus empirically select the ensemble number $E_c$, which is illustrated in Sec. 5.2.1. The selected members are further fine-tuned with new classification heads to the current task.

### 3.3 INTRA-TASK MODEL ENSEMBLE

Our filter decomposition not only allows ensemble with past members, but also permits an efficient way to create ensemble members within a task. In the regular deep ensemble scenario (Lakshminarayanan et al., 2017), different members are instantiated as multiple CNNs, and need to be learned separately to ensure independence among members. Plus, obtaining ensemble results also requires inferences with multiple CNNs. This introduces significant cost in both time and memory, making it inappropriate in continual learning settings. The proposed atom decomposition allows a new way of parameterization of ensemble members within a task to improve performance while substantially reducing the training and testing time cost. Formally, given task $t$, the intra-task ensemble model $\mathcal{F}^t_{w-ens}$ is composed by $\{\hat{\mathcal{F}}^{t,1}, ..., \hat{\mathcal{F}}^{t,E_w}\}$, where $E_w$ is the number of models. Rather than instantiating them as different CNN models, we reparameterize them with member-specific atoms $\{\hat{\mathbf{D}}^{t,1}, ..., \hat{\mathbf{D}}^{t,E_w}\}$, and member-shared coefficient $\boldsymbol{\alpha}^t$, as shown in Fig. 3. In this way, we create multiple virtual members using a single CNN model by simply maintaining different groups of filter atoms in each convolutional layer. The forward pass of $\mathcal{F}^t_{w-ens}$ can then be conducted by group convolution,

$$\mathcal{F}^t_{w-ens}(\mathbf{x}^t) = \frac{1}{E_w} \sum_{i=1}^{E_w} \hat{\mathcal{F}}^{t,i}(\mathbf{x}^t) = \mathcal{F}^t_{w-ens}(\hat{\mathbf{x}}^t, \hat{W}^t_{w-ens}), \tag{10}$$

where $\hat{\mathbf{x}}^t \in \mathbb{R}^{n \times (c \times E_w) \times h \times w}$ is the input repeated by $E_w$ times, $\hat{W}^t_{w-ens} = [\boldsymbol{\alpha} \times \hat{\mathbf{D}}^{t,1} \mid ... \mid \boldsymbol{\alpha} \times \hat{\mathbf{D}}^{t,E_w}] \in \mathbb{R}^{c \times (c' \times E_w) \times k \times k}$ ($\mid$ denotes concatenation) is the filter for group convolution that is concatenated from per-member reconstructed filters. To enforce the independence of different members, we initialize member-specific atoms separately before training.

With diverse predictions from different virtual members, our intra-task ensemble can directly boost the performance in task-incremental settings. Furthermore, our intra-task model ensemble makes the model to better distinguish data from out-of-task distribution, i.e., $\mathbf{x}^p(p \neq t)$, and shows high entropy in its predictive distribution which is particularly useful in class-incremental (CI) setting. Note that in CI we perform task prediction first, select a specific group of atoms, and then perform classification within task. Task id is selected based on minimal-entropy criterion on predictive distributions. Thus, intra-task ensemble can enhance the accuracy of task prediction, and then the overall CI performance.

## 4 EXPERIMENTAL SETUP

**Datasets.** We first validate our method with the Class-Incremental (CI) setting with CIFAR100 and ImageNet-Subset, which contains 100 classes selected from ImageNet (with random seed 1993). For each dataset, half of classes are used for model learning in the 0-th task. Then the remaining classes are equally split into N tasks, where N can be 5 or 10. Note that in the CI setting, task information is not provided during testing. Then, we report the performance of our method under the Task-Incremental (TI) setting. We validate our approach on *5-Split MNIST* (LeCun et al., 1998), *20-Split CIFAR100* (Krizhevsky et al., 2009), and *20-Split miniImageNet* (Vinyals et al., 2016). The *5-Split MNIST* uniformly splits the original 10 classes of 0-9 MNIST digits into 5 sequential tasks. *20-Split CIFAR100* and *20-Split miniImageNet* are both constructed by randomly splitting 100 classes into 20 tasks with 5 classes per task. Details of each dataset are provided in the Appendix C.

Table 1: Class-incremental results on CIFAR100 and ImageNet-Subset with different number of incremental tasks (N). We report average incremental accuracy for all methods.

| Method | CIFAR-100 | | | ImageNet-Sub | | |
| --- | --- | --- | --- | --- | --- | --- |
| | Acc.(N=5) | Acc.(N=10) | Memory | Acc.(N=5) | Acc.(N=10) | Memory |
| LwF-E (Li & Hoiem, 2017) | 57.03 | 56.82 | | 65.51 | 65.58 | |
| EWC-E (Kirkpatrick et al., 2017a) | 56.28 | 55.41 | | 65.22 | 64.13 | |
| iCaRL (Rebuffi et al., 2017) | 57.17 | 52.57 | 6.2 MB | 65.04 | 59.53 | 301.4 MB |
| SDC (Yu et al., 2020) | 57.10 | 56.80 | | 65.60 | 65.70 | |
| BiC (Wu et al., 2019) | 59.36 | 54.20 | | 70.07 | 64.96 | |
| LUCIR (Hou et al., 2019) | 63.12 | 60.14 | | 70.47 | 68.09 | |
| Ours (Base) | 60.23 | 55.54 | **0.2 MB** | 71.74 | 64.40 | **2.7 MB** |
| **w/ $E_w = 2$** | **65.44** | **62.48** | **0.5 MB** | **75.85** | **72.11** | **7.7 MB** |

**Network architectures and implementation details.** For the class-incremental setting, we utilize ResNet-32 for CIFAR100 and ResNet-18 for ImageNet-Subset as (Rebuffi et al., 2017). For the task-incremental setting, we adopt an AlexNet-like network. Note that we substitute all convolutional layers in both models with our decomposed version. Details of architecture are shown in the Appendix C. In terms of the proposed ensemble strategies, inter-task ensemble is only deployed in the task-incremental setting as task IDs are needed, while intra-task ensemble is utilized in both task-incremental and class-incremental settings. In inter-task ensemble, we set $M = 3$ for the dimension of filter subspaces. When intra-task ensemble and inter-task ensemble are adopted together, we use the member with the best results in every task for calculating task relevancy. We provide the ablation study for the inter-class and intra-class ensemble numbers $E_c$ and $E_w$ in Sec. 5.2.1. Training details are showed in the Appendix C.3. For forward knowledge transfer, the most recent atoms are used to initialize the atoms of the current task.

**Evaluation metrics.** In the class-incremental setting, we evaluate the model's average class-incremental accuracy. In the task-incremental setting, we measure the performances with ACC as the average test accuracy across all tasks. To measure the forgetting, we adopt the backward transfer, BWT, which shows how the previous tasks performance has degraded due to learning new tasks. Details of these two measurements are provided in Appendix C.4.

## 5 RESULTS AND DISCUSSION

In this section, we start from the challenging class-incremental setting. Then we move to task-incremental setting with self-comparison experiments to validate the effectiveness of some key ingredients of our methods, and show the results of the proposed method on several real-world datasets. In both settings, our method achieves improvements over state-of-the-art methods with significant less memory usage.

### 5.1 CLASS-INCREMENTAL EXPERIMENTS

As a more challenging setting, class-incremental (CI) learning does not provide task id during testing. As mentioned in Sec. 3.3, we handle this setting by breaking it down to a two-level task, task prediction based on minimal-entropy criterion, and then within-task classification. In CI experiments, we select the number of atoms $m = 12$ and the number of members for intra-task ensemble $E_w = 2$. We benchmark our method by comparing with many existing methods that store 2000 exemplars in the external memory. As shown in Tab. 1, our method with intra-task ensemble achieves the best results with an order of magnitude less memory usage on both CIFAR100 and ImageNet-Subset, which validates the effectiveness and scalability of our framework in the challenging CI setting.

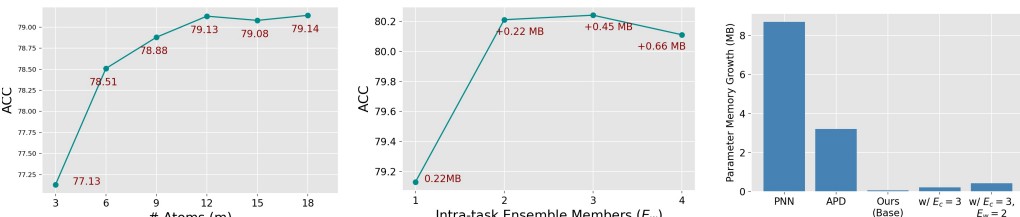

Figure 4: **Left & Middle:** Ablation study on the number of atoms ($m$) and intra-task ensemble members ($E_w$) on 20-Split CIFAR100. **Right:** Parameter memory growth per-task for 20-Split CIFAR100. The proposed method shows significantly lower memory growth than other expansion methods.

Table 2: Results on *20-Split CIFAR100* and *20-Split miniImageNet*. (*) We re-implement PNN and APD with our network architecture. Analysis on memory is provided in Appendix C.10.

| Method | CIFAR-100 | | | miniImageNet | | |
|---|---|---|---|---|---|---|
| | ACC% | BWT% | Memory (MB) | ACC% | BWT% | Memory (MB) |
| EWC (Kirkpatrick et al., 2017a) | 55.60± 1.11 | 23.53± 1.19 | - | 36.61 | 28.17 | - |
| HAT (Serra et al., 2018) | 76.96± 1.23 | 0.01± 0.02 | - | 59.45 | 0.00 | - |
| PNN* (Rusu et al., 2016) | 82.25± 0.04 | 0.00± 0.00 | 165.3 | 70.96 | 0.00 | 165.3 |
| APD* (Yoon et al., 2019) | 77.03± 0.14 | -0.02± 0.01 | 60.5 | 61.67 | 0.07 | 60.5 |
| iCaRL (Rebuffi et al., 2017) | 58.08± 1.44 | 24.22± 1.35 | 28.8 | - | - | 173.6 |
| A-GEM (Chaudhry et al., 2018c) | 54.38± 3.84 | -21.99± 4.05 | 16 | 52.43 | -15.23 | 110.1 |
| ER-RES (Chaudhry et al., 2019a) | 66.78± 0.48 | -15.01± 1.11 | 16 | 57.32 | -11.34 | 110.1 |
| GCL (Tang & Matteson, 2020) | 74.51± 0.99 | 6.54± 1.26 | 7.2 | 61.54 | 6.10 | 43.4 |
| ACL (Ebrahimi et al., 2020) | 78.08± 1.25 | 0.00± 0.01 | - | 62.07 | 0.00 | 8.5 |
| Ours (Base) | 79.13± 0.12 | 0.00± 0.00 | 0.14 | 66.01 | 0.00 | 0.14 |
| w/ $E_c = 3$ | 79.91± 0.15 | 0.00± 0.00 | 0.43 | 66.83 | 0.00 | 0.43 |
| w/ $E_w = 2$. | 80.21± 0.21 | 0.00± 0.00 | 0.28 | 67.29 | 0.00 | 0.28 |
| **w/ both** | **80.75**± 0.18 | **0.00**± 0.00 | **0.86** | **67.84** | **0.00** | **0.86** |

## 5.2 RESULTS ON TASK-INCREMENTAL SETTINGS

### 5.2.1 SELF COMPARISONS ON *20-Split CIFAR100*

In this section, we empirically analyze inter-task ensemble, intra-task ensemble, and selection of atoms of our method in the task-incremental setting. We analyze the performance of our base model with different number of atoms $m$, As shown in the left of Fig. 4, $m = 12$ is the best choice in terms of both performance and efficiency. We then test the task relevancy assessment based on subspace distance, and its instructive effect to inter-task ensemble. As shown in Fig. 5, the past models with small Grassmann distance to the current model lead to performance improvement by model ensemble. And past models with large Grassmann distance to the current one, in fact result in degraded performances. In general, we test the ACC of model ensemble with the top-1 to top-3 relevant model as well as the most irrelevant model starting from the 5-th task. As shown in Fig. 5, model ensemble with top-3 relevant models achieves the best results, and thus we set $E_c = 3$ in subsequent experiments. We then explore intra-task ensemble with different members $E_w = 2, 3, 4$. As illustrated in the middle of Fig. 4, ensemble within task enhances the performance consistently, and we choose $E_w = 2$ for the best performance efficiency trade-off. We further illustrate the scalability of the proposed method on the right of Fig. 4. Compared to PNN (Rusu et al., 2016) and APD (Yoon et al., 2019), the size of our method scales much slower with the number of tasks, even with ensemble adopted.

### 5.2.2 COMPARISONS WITH BENCHMARKS

We further report our results on *20-Split miniImageNet* and *20-Split CIFAR100* in Tab. 2. Comparing with regularization-based, memory-based, and expansion-based models, the proposed method achieves the best results even with the base model. Adopting inter-task ensemble with the top-3 relevant past models, along with intra-task ensemble with 2 members, our method achieves further improved results. Especially in *20-Split miniImageNet*, the previous state-of-the-art method, ACL (Ebrahimi et al., 2020) achieves ACC of 62.07 with a memory in size of 8.5 MB; whereas the proposed method improves the results significantly to 67.84 with a merely 0.86 MB memory. When comparing with other memory-based methods besides ACL, the proposed method demonstrates superior scalability reflected by the much smaller memory size. We provide additional results on the standard *5-Split MNIST* dataset with our base model. As shown in Tab. D, our base method outperforms both regularization-based methods and memory-based methods in terms of ACC with

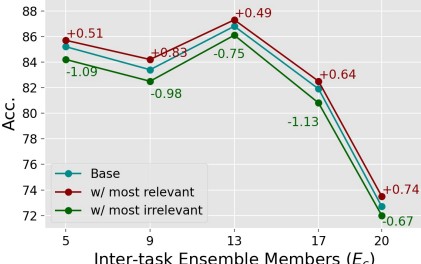

| Method | ACC | Avg. $d_k$ | Memory (MB) |
|---|---|---|---|
| Base | 79.13 | - | 0.14 |
| w/ Ens. top-1 | 79.61 (+0.48) | 0.43 | 0.23 |
| w/ Ens. top-2 | 79.82 (+0.69) | 0.48 | 0.34 |
| w/ Ens. top-3 | **79.91 (+0.78)** | 0.56 | 0.43 |
| w/ Ens. last-1 | 78.27 (-0.86) | 1.21 | 0.23 |

Figure 5: (Plot) Ensemble effect of the base model with the most relevant and irrelevant past model. (Table) Ablation studies on number of ensemble selections $n$.

much smaller memory. The improvements over the state-of-the-art methods and the outstanding scalability validate the effectiveness of our method on solving real-world continual learning problems.

## 6 RELATED WORK

**Continual learning.** Recent advances on continual learning are driven by three main directions, regularization-based, memory-based and expansion-based methods. (Kirkpatrick et al., 2017b; Aljundi et al., 2018a; Lee et al., 2017; Zenke et al., 2017b; Kolouri et al., 2019) determine the importance of each model's parameter per task, which prevents the important parameters from being updated for new tasks. For example, (Kirkpatrick et al., 2017b) specify the performance of each weight with the Fisher information matrix. Theses methods can be naturally explored from the lens of Bayesian optimization (Nguyen et al., 2018; Titsias et al., 2020; Schwarz et al., 2018; Ebrahimi et al., 2019; Ritter et al., 2018). All these methods address catastrophic forgetting by adding regularization terms. As pointed out in (De Lange et al., 2019), the penalty term proposed in such methods are unable to prevent drifts in the loss landscape of previous tasks. While alleviating forgetting, the penalty also unavoidably prevents the plasticity to absorb new information from future tasks learned over a long timescale (Hadsell et al., 2020).

(De Lange et al., 2019) assumes it is feasible to access data from previous tasks by having a fixed-size memory or a generative model able to produces samples from previous tasks (Lopez-Paz & Ranzato, 2017; Riemer et al., 2018; Rios & Itti, 2018; Shin et al., 2017). (Rebuffi et al., 2017) introduces models augmented with fixed-size memory, which accumulates samples in the proximity of class centers. (Chaudhry et al., 2019b) proposes another memory-based model by exploiting a reservoir sampling strategy in the raw input data selection phase. Rather than storing the original samples, (Chaudhry et al., 2018a) accumulates the parameter gradients during task learning. (Shin et al., 2017) incorporate a generative model into a continual learning model to alleviate catastrophic forgetting by producing samples from previous task and retraining the model using data from previous tasks and the the current one. These methods assume an extra neural network, such as a generative model or a memory. Different from replay-based methods, which benefit from a memory to retrain their model over previous tasks, our method requires storage of tiny atoms for each previous task only, which is more scalable and do not suffer from potential forgetting caused by the inconsistent memory reply in generative-based methods.

(Rusu et al., 2016; Yoon et al., 2018; Jerfel et al., 2019; Li et al., 2019) allocate a subset of the model parameters for each task. Model expansion can be achieved by a gating mechanism (Wortsman et al., 2020; Masse et al., 2018), or by incrementally adding new parameters to the models (Rusu et al., 2016). Incrementally learning and pruning provides another direction (Mallya & Lazebnik, 2018). Given an over-parametrized model with the ability to learn potentially many tasks, (Mallya & Lazebnik, 2018) achieves model expansion by pruning the parameters not contributing to the performance of the current task, while keeping them avail- able for future tasks. Comparing to the aforementioned methods, the proposed method provides a filter subspace view of modeling multiple tasks, which further allows two kinds of model ensemble .

**Filter atom decomposition.** (Qiu et al., 2018) proposes an convolutional filter decomposition as a truncated expansion with pre-fixed filter atoms. It not only reduces the number of learnable parameters, but also imposes filter regularity with the usage of Fourier-Bessel basis. This work further inspires other works in domain adaptation (Wang et al., 2020b), adaptive convolution (Wang et al., 2021b; 2020a), image generation (Wang et al., 2021a; 2019), video understanding (Miao et al., 2021), rotation equivariance (Cheng et al., 2018) and graph convolution (Cheng et al., 2020).

## 7 CONCLUSION

In this paper, motivated by the task subspace modeling literature, we enforced a low-rank filter structure to each CNN layer across tasks in continual learning. By performing atom-coefficient filter decomposition, we learned for each task a new filter subspace for each layer, while keeping subspace coefficients shared across tasks. This simple method allows highly efficient model storage and retrieval using a small footprint atom memory. The proposed method provided a guarantee against forgetting, and we demonstrated further performance improvements through model ensemble. The performance was evaluated on various continual learning tasks, and the effectiveness and scalability were demonstrated by the state-of-the-art accuracy and the tiny size of atom memory.

ACKNOWLEDGEMENT

This work is supported by DARPA TAMI program under No. HR00112190038.

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

## APPENDIX

## A  EXPECTED EXCESS RISK

**Problem setup:**  Suppose we have $T$ tasks sequentially. The data distribution $\mathcal{D}^t = (\mathcal{X}^t, \mathcal{Y}^t)$ associates $t$-th task ($t \in \{1, 2, ..., T\}$). For each task $t \in [T]$, we assume $\phi^t : \mathcal{X} \to Z$ is a representation function, which maps input images to feature space $Z \subseteq \mathbb{R}^k$. These representation functions are restricted in function class $\Phi$, such as neural networks. There is also a predictor $\mathbf{w}^t : Z \to \mathcal{Y}$ for task $t \in [T]$ mapping the feature representation to labels. Our object is to minimize empirical risk:

$$\mathbb{E}_{(\mathcal{X}^t, \mathcal{Y}^t)}[\mathcal{L}(\mathcal{F}^t(\mathcal{X}^t; \theta), \mathcal{Y}^t)].$$

For analysis purpose, we consider representation function $\phi^t$ as single convolutional layer, which can be decomposed as $\phi^t = \boldsymbol{\alpha} \times \mathbf{D}^t$. With the predictor $\mathbf{w}^t$, the loss can be written as:

$$\mathcal{L}(\mathcal{F}^t(\mathcal{X}^t; \theta), \mathcal{Y}^t) = \frac{1}{2N^t} \sum_{i=1}^{N^t} (\mathbf{y}_i - \langle \mathbf{w}^t, (\boldsymbol{\alpha} \times \mathbf{D}^t(\mathbf{x}_i))_+ \rangle)^2,$$

where $\mathbf{D}^t(\mathbf{x})$ is the convolution operation on the input images, $(\cdot)_+$ is ReLU activation. As the input $\mathbf{x} \in \mathbb{R}^{c \times h \times w}$ and atoms $\mathbf{D} \in \mathbb{R}^{m \times k \times k}$, where $h \times w$ are the size of input images, $c$ is the number of input channel, and $m$ is the number of kernels which have size of $k \times k$. $\mathbf{D}^t(\mathbf{x})$ contains $c \times m$ convolution operations that maps input $\mathbf{x}$ into the latent $z_l \in \mathbb{R}^{cm \times h' \times w'}$. The latent $z_l$ then maps into feature space $Z$ by $\boldsymbol{\alpha} \times z_l$.

We can treat each convolution operation as matrix product by converting the convolution kernel to a doubly blocked Toeplitz matrix $\mathbf{D}'^t \in \mathbb{R}^{hw \times h'w'}$ (Gray, 2006), and converting input image $\mathbf{x}$ into $\mathbf{x}' \in \mathbb{R}^{c \times hw}$. Therefore, the convolution operation $\mathbf{D}^t(\mathbf{x})$ is transformed as reshaped image $\mathbf{x}'$ multiple by $m$ reformed kernels $\mathbf{D}'^t$. To simplify the problem, we assume consider each channel of the input images and each base of atoms, such that $c = 1$ and $m = 1$. Consider of all $N^t$ samples $X^t \in \mathbb{R}^{N^t \times hw}$ with labels $Y^t \in \mathbb{R}^{N^t}$ for task $t$, Thus, the loss can be written as:

$$\mathcal{L}(\mathcal{F}^t(\mathcal{X}^t; \theta), \mathcal{Y}^t) = \frac{1}{2N^t} \sum_{i=1}^{N^t} (\mathbf{y}_i - \langle \mathbf{w}^t, (\boldsymbol{\alpha} \times \mathbf{D}^t(\mathbf{x}_i))_+ \rangle)^2$$

$$= \frac{1}{2N^t} \| Y^t - (\boldsymbol{\alpha}^\intercal X^t \mathbf{D}'^t)_+ \mathbf{w}^{\intercal t} \|^2$$

The first task trains the $\boldsymbol{\alpha}$, $\mathbf{D}'^1$, and $\mathbf{w}^1$ at the same time, but the following tasks only optimize over $\mathbf{D}^t$ and $\mathbf{w}^t$ and keep $\boldsymbol{\alpha}$ fixed. With fixed $\boldsymbol{\alpha}$, we can represent the samples as $\tilde{X}^t = \boldsymbol{\alpha}^\intercal X^t$. By solving the object, we get $(\hat{\mathbf{D}}^t, \hat{\mathbf{w}}^t)$. We also assume that there is an optimal atom $\mathbf{D}^{t,*}$ of representation function $\phi^{t,*}$ and prediction function $\mathbf{w}^{t,*}$ for task $t$. We now analyze the bound for the excess risk of our learned model on the task $t$, *i.e.*, how much our learned model $(\hat{\mathbf{D}}^t, \hat{\mathbf{w}}^t)$ performs worse than the optimal model $(\mathbf{D}^{t,*}, \mathbf{w}^{t,*})$ on the task $t$:

$$\begin{aligned}
ER(\hat{\phi}^t, \hat{\mathbf{w}}^t) &= L_{\mathcal{D}^t}(\hat{\mathbf{D}}^t, \hat{\mathbf{w}}^t) - L_{\mathcal{D}^t}(\mathbf{D}^{t,*}, \mathbf{w}^{t,*}) \\
&= \frac{1}{2}((\tilde{X}^t \hat{\mathbf{D}}'^{\intercal t})_+ \hat{\mathbf{w}}^{\intercal t} - (\tilde{X}^t \mathbf{D}'^{\intercal t,*})_+ \mathbf{w}^{\intercal t,*})^2
\end{aligned} \quad (1)$$

A standard lifting of neural networks can be formulated as infinite dimension linear regression (Du et al., 2020). Define the infinite feature vector with the coordinates $\phi(x)_b = (b^\intercal x)_+$ for every $b \in \mathbb{S}^{d_0 - 1}$. Let $\beta_t$ be a signed measure on $\mathbb{S}^{d_0 - 1}$. The inner product notation denotes integration: $\beta^\intercal \phi(x) = \int_{\mathbb{S}^{d_0-1}} \phi(x)_b d\beta_b$. By convert input feature $x$ to the lifted feature vector $\phi(x)$, according to (Du et al., 2020), Equation. 1 can be bounded by: $R(\hat{\phi}^t, \hat{\mathbf{w}}^t) = \frac{1}{2}(\tilde{X}^t \hat{\mathbf{D}}'^{\intercal t} \hat{\mathbf{w}}^{\intercal t} - \tilde{X}^t \mathbf{D}'^{\intercal t,*} \mathbf{w}^{\intercal t,*})^2$.

Let $\mathbb{E}[\tilde{\mathbf{x}} \tilde{\mathbf{x}}^\intercal] = \tilde{\Sigma}$, we have

$$\frac{1}{2}(\tilde{X}^t \hat{\mathbf{D}}'^{\intercal t} \hat{\mathbf{w}}^{\intercal t} - \tilde{X}^t \mathbf{D}'^{\intercal t,*} \mathbf{w}^{\intercal t,*})^2 = \frac{1}{2} \| \tilde{\Sigma}^{1/2} \mathbf{D}'^{\intercal t,*} \mathbf{w}^{\intercal t,*} - \tilde{\Sigma}^{1/2} \hat{\mathbf{D}}'^{\intercal t} \hat{\mathbf{w}}^{\intercal t} \|^2.$$

With Assumption 1 and 2, based on the inequality in (Du et al., 2020), for a fixed $\delta > 0$, we further have

$$\frac{1}{2} \| \tilde{\Sigma}^{1/2} \mathbf{D}'^{\intercal t,*} \mathbf{w}^{\intercal t,*} - \tilde{\Sigma}^{1/2} \hat{\mathbf{D}}'^{\intercal t} \hat{\mathbf{w}}^{\intercal t} \|^2 \leq \epsilon_{ee,1}^2 \bar{R} + \epsilon_{ic,1}^2 \bar{R}^2 + \epsilon_{ee,2}^2 r + \epsilon_{ic,2}^2 r^2,$$

where $\epsilon_{ee,1}^2 = \frac{1}{\sqrt{N^1}} \sigma (log \frac{1}{\delta})^{3/2} log(n) \sqrt{\| \tilde{\Sigma} \| + Tr(\tilde{\Sigma})}$, $\epsilon_{ee,2}^2 = \frac{1}{\sqrt{N^t}} \sigma (log \frac{1}{\delta})^{3/2} log(n) \sqrt{\| \tilde{\Sigma} \|}$ are estimation error and $\epsilon_{ic,1} = \frac{2C\rho^2}{\sqrt{N^1}} (\sqrt{Tr(\tilde{\Sigma})} + \sqrt{log \frac{2}{\delta} \| \tilde{\Sigma} \|})$, $\epsilon_{ic,2} = \frac{C\rho^2}{\sqrt{N^t}} \sqrt{\| \tilde{\Sigma} \| \bar{R} log \frac{1}{\delta}}$ are intrinsic dimension concentration error, $\bar{R} = \frac{1}{2} \| \boldsymbol{\alpha}^* \times \mathbf{D}^{t,*} \|_F^2 + \frac{1}{2} \| \mathbf{w}^{t,*} \|_F^2$. Therefore, suppose we have the same number of samples

in each task, with probability at least $1 - \delta$ over the samples, the expected excess risk of the learned atom and predictor on the task satisfies:

$$\mathbb{E}[ER(\hat{\phi}^t, \hat{\mathbf{w}}^t)] \leq \sigma \bar{R} \cdot \tilde{O}(\frac{\sqrt{Tr(\Sigma)} + \sqrt{\|\Sigma\|_2}}{\sqrt{N^t}}) + \rho^4 \bar{R}^2 \cdot \tilde{O}(\frac{Tr(\Sigma) + \|\Sigma\|_2}{N^t})$$

## B  ALGORITHM

We provide the algorithm of the proposed method in Alg. 1.

---

**Algorithm 1** Continual Learning with Filter Atom Swapping

---

Initialize $\mathcal{M}_{\mathbf{D}} \leftarrow [\quad], \boldsymbol{\alpha} = \{\boldsymbol{\alpha}_i\}_{i=1}^l \leftarrow \boldsymbol{\alpha}^0, \mathbf{D}^1 = \{\mathbf{D}_i^1\}_{i=1}^l \leftarrow \mathbf{D}^{1,0}$.
**for** task $t = 1, 2, ..., T$ **do**
  **if** $t == 1$ **then**
    Optimize $\boldsymbol{\alpha}, \mathbf{D}^1$ according to (3).
    Update Atom Memory $\mathcal{M}_{\mathbf{D}} \leftarrow [\mathcal{M}_{\mathbf{D}}, \mathbf{D}^1]$.
  **else**
    **if** $t < 5$ **then**
      Initialize $\mathbf{D}^t = \{\mathbf{D}_i^t\}_{i=1}^l \leftarrow \mathbf{D}^{t,0}$, then optimize $\mathbf{D}^t$ according to (2).
      Update Atom Memory $\mathcal{M}_{\mathbf{D}} \leftarrow [\mathcal{M}_{\mathbf{D}}, \mathbf{D}^t]$.
    **else**
      Initialize $\mathbf{D}^t = \{\mathbf{D}_i^t\}_{i=1}^l \leftarrow \mathbf{D}^{t,0}$, then optimize $\mathbf{D}^t$ according to (2).
      Initialize distance list $\mathbf{d}^t = [0, ..., 0]$, where length($\mathbf{d}^t$) $= t - 1$.
      **for** $j$=1:t-1 **do**
        Calculate $d_M(\mathbf{D}_l^t, \mathbf{D}_l^j)$ in the last convolutional layer according to (8), and update $\mathbf{d}^t[j] = d_M(\mathbf{D}_l^t, \mathbf{D}_l^j)$
      **end for**
      Sort $\mathbf{d}^t$ in ascending order, and select least $E_c$ indices to construct $\mathcal{S}^t$.
      Finetune additional $n$ heads on current task.
      Update Atom Memory $\mathcal{M}_{\mathbf{D}} \leftarrow [\mathcal{M}_{\mathbf{D}}, \mathbf{D}^t]$.
    **end if**
  **end if**
**end for**
**return** $\mathcal{M}_{\mathbf{D}}, \boldsymbol{\alpha}$.

---

## C  EXPERIMENTAL DETAILS

### C.1  DATASET STATISTICS

We provide the dataset statistics of *10-Split CIFAR100* used in class-incremental setting in Tab. A, and statistics of 20-Split CIFAR100and 20-Split miniImageNetin task-incremental setting in Tab. B and Tab. C.

Table A: Statistics of *10-Split CIFAR100*.

|  | 10-Split CIFAR100 |
|---|---|
| # tasks | 10 |
| Img. Size | $32 \times 32 \times 3$ |
| # tasks/task | 10 |
| # Training samples/task | 4500 |
| # Validation samples/task | 500 |
| # Test samples/task | 100 |

### C.2  NETWORK ARCHITECTURE

#### C.2.1  ARCHITECTURE FOR CLASS-INCREMENTAL LEARNING

We adopt the ResNet-32 as in (Rebuffi et al., 2017), which includes 31 convolutional layers and 1 fully connected layer. We substitute every convolutional layer with the one has the proposed decomposed filters ($m = 12$) that has the same number of input/ output channels.

Table B: Statistics of 20-Split CIFAR100and 20-Split miniImageNet.

|  | 20-Split CIFAR100 | 20-Split miniImageNet |
|---|---|---|
| # tasks | 20 | 20 |
| Img. Size | $32 \times 32 \times 3$ | $84 \times 84 \times 3$ |
| # tasks/task | 5 | 5 |
| # Training samples/task | 2,125 | 2,125 |
| # Validation samples/task | 375 | 375 |
| # Test samples/task | 500 | 500 |

Table C: Statistics of 5-Split MNIST.

| Task | (0,1) | (2,3) | (4,5) | (6,7) | (8,9) |
|---|---|---|---|---|---|
| # Training samples/task | 10,766 | 10,276 | 9,574 | 10,356 | 10,030 |
| # Validation samples/task | 1,899 | 1,813 | 1,689 | 1,827 | 1,770 |
| # Test samples/task | 2,115 | 2,042 | 1,874 | 1,986 | 1,983 |

### C.2.2 ARCHITECTURE FOR TASK-INCREMENTAL LEARNING

**Architecture for 5-Split MNIST.** Conv($k$=3, $m$=12, 16)-ReLU-Dropout(0.2)-MaxPool(2)-Conv($k$=3, $m$=12,32)-ReLU-Dropout(0.2)-MaxPool(2)-Conv($k$=3, $m$=12, 64)-ReLU-Dropout(0.2)-GAP-FC(32)-ReLU-Dropout(0.5)-FC(2),

where GAP stands for global average poling, Conv($k$, $m$, $c$) means the convolution layer with kernel size $k$, $m$ atoms, $c$ output channels, Dropout($p$) indicates the dropout layer with probability $p$, and FC($c$) is the fully-connect layer with $c$ output channels.

**Architecture for 20-Split CIFAR100and 20-Split miniImageNet.** Conv(k=3, m=12, 64)-ReLU-MaxPool(2)-Conv(k=3, m=12, 192)-ReLU-MaxPool(2)-Conv(k=3, m=12, 384)-ReLU-Conv(k=3, m=12, 256)-ReLU-Conv(k=3, m=12, 256)-ReLU-MaxPool(2)-GAP-Dropout(0.5)-FC(5)

### C.3 TRAINING DETAILS

### C.3.1 CLASS-INCREMENTAL LEARNING

For CIFAR100, we choose SGD with batch-size of 128, learning rate of 0.01, momentum of 0.9 and weight decay 1e-3. The model is trained for 250 epochs, with learning rate drop by 0.1 at the 100-th and 200-th epoch. For ImageNet-Subset, we choose SGD with batch-size of 128, learning rate of 0.05, momentum of 0.9, and weight decay 1e-4. The model is trained for 150 epochs, with learning rate drop by 0.1 at the 90-th and 120-th epoch. For CUBS and Flowers, We choose SGD with momentum 0.9, batch-size of 64, and weight decay of 1e-4. On CUBS, the model is finetuned for 200 epochs with learning rate of 1e-2 (drop by 0.1 at the 100th and 160th epoch) on the first task, and trained for 120 epochs with learning rate of 5e-3 (drop by 0.1 at the 70th epoch) on task 2-6. On Flowers, the model is finetuned for 120 epochs with learning rate of 1e-2 (decay by 0.1 at the 50th and 90th epoch) on the first task, and trained for 70 epochs with learning rate of 1e-3 (decay by 0.1 at the 40th epoch) on task 2-6.

### C.3.2 TASK-INCREMENTAL LEARNING

We choose SGD with batch-size of 64 for all experiments under the task-incremental setting. For 5-Split MNIST, we train the model for 20 epochs with learning rate of 0.001. For 20-Split CIFAR100, we train the model for 100 epochs with learning rate of 0.001, which drops by 0.1 at the 60-th epoch. For 20-Split miniImageNet, we train the model for 200 epochs with learning rate of 0.001, which drops at the 100-th and 150-th epoch.

### C.4 METRIC

ACC and BWT used in the task-incremental settings can be expressed as,

$$\text{ACC} = \frac{1}{T} \sum_{i=1}^{T} \text{Acc}_{T,i}, \quad \text{BWT} = \frac{1}{T-1} \sum_{i=1}^{T-1} \text{Acc}_{T,i} - \text{Acc}_{i,i}, \tag{2}$$

where $T$ is total number of tasks, $\text{Acc}_{i,j}$ is performance on $j$-th task after learning $i$-th task (Lopez-Paz & Ranzato, 2017).

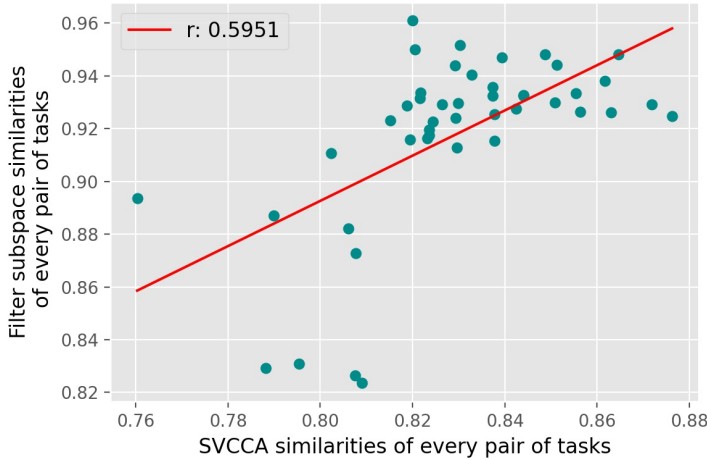

Figure A: Correlation analysis between task similarities calculated with CCA (Raghu et al., 2017) and task similarities assessed with filter subspace distance.

## C.5 CORRELATION ANALYSIS BETWEEN CCA AND FILTER SUBSPACE DISTANCE.

We conduct the analysis between the task similarities measured with CCA (Raghu et al., 2017) on deep feature spaces, and the proposed task similarities assessed with Grassmann distance on filter subspaces. The analysis is conducted on 20-Split CIFAR100 with the proposed model. We randomly select 50 task pairs from $\binom{20}{2}$ task pairs. Given a task pair, we use 500 images from each task to calculate the CCA similarity, and use the inverse of their Grassmann distance between their last layers' filter subspaces as the filter-subspace similarity. As shown in Fig. A, filter-subspace similarities quite correlates to CCA similarities which we deem as the golden standard, while they are significantly more efficient to calculate with our atom continual learning framework.

## C.6 ADDITIONAL TASK-INCREMENTAL EXPERIMENTAL RESULTS

We provide additional results on the standard *5-Split MNIST* dataset. Note that due to the simplicity of *5-Split MNIST*, we only apply on it our base model, which is composed by cross-task coefficient sharing and atom swapping only. As shown in Tab. D, our base method outperforms both regularization-based methods and memory-based methods in terms of ACC. Besides, the proposed method demonstrates a guarantee against forgetting reflected by 0 in BWT. Note that the proposed base model is much smaller in size compared to the other methods. Moreover, comparing to GEM Lopez-Paz & Ranzato (2017) and VCL Nguyen et al. (2018), our base network adopts far smaller memory for storing merely filter atoms for 3 layers, each of them has a hundred of parameters only.

Table D: Results on *5-Split MNIST*.

| Method | ACC% | BWT% | Arch. Size (MB) | Memory (MB) |
|---|---|---|---|---|
| EWC (Kirkpatrick et al., 2017a) | 95.78± 0.35 | -4.20± 0.21 | 1.1 | - |
| HAT (Serra et al., 2018) | 99.59± 0.01 | 0.00± 0.04 | 1.1 | - |
| UCB (Ebrahimi et al., 2019) | 99.63± 0.02 | 0.00± 0.00 | 2.2 | - |
| VCL (Nguyen et al., 2018) | 95.97± 1.03 | -4.62± 1.28 | 1.1 | - |
| GEM (Lopez-Paz & Ranzato, 2017) | 94.34± 0.82 | -2.01± 0.05 | 6.5 | 0.63 |
| VCL-C (Nguyen et al., 2018) | 93.6± 0.2 | -3.10± 0.20 | 1.7 | 0.63 |
| ACL (Ebrahimi et al., 2020) | 99.76± 0.03 | 0.01± 0.01 | 1.6 | - |
| MIR (Aljundi et al., 2019a) | 87.60± 0.70 | 7.00± 0.90 | 5.9 | 0.71 |
| GSS (Aljundi et al., 2019b) | 84.80± 1.80 | - | 3.7 | 0.58 |
| **Ours (Base)** | **99.84± 0.05** | **0.00± 0.00** | **0.24** | **0.04** |

## C.7    ADDITIONAL CLASS-INCREMENTAL RESULTS

### C.7.1    ADDITIONAL RESULTS ON *10-Split CIFAR100*

We provide results *10-Split CIFAR100* under class-incremental setting with the ResNet-18 (He et al., 2016) in Tab. E and Fig. B. Our method achieves the best result with a large margin.

Table E: Class-incremental results on *10-Split CIFAR100*.

| Method | Acc. (avg.) | Acc. (last) |
|---|---|---|
| LwF (Li & Hoiem, 2017) | 44.5 | 23.9 |
| EWC (Kirkpatrick et al., 2017a) | 36.2 | 16.4 |
| SI (Zenke et al., 2017a) | 37.8 | 23.3 |
| MAS (Aljundi et al., 2018b) | 33.4 | 15.4 |
| RWalk (Chaudhry et al., 2018b) | 35.2 | 17.9 |
| DMC (Zhang et al., 2020) | 57.1 | 36.2 |
| Ours (Base) | 58.6 | 40.57 |
| w/ $E = 2$ | **62.4** | **46.37** |
| w/ $E = 3$ | **62.9** | **46.50** |

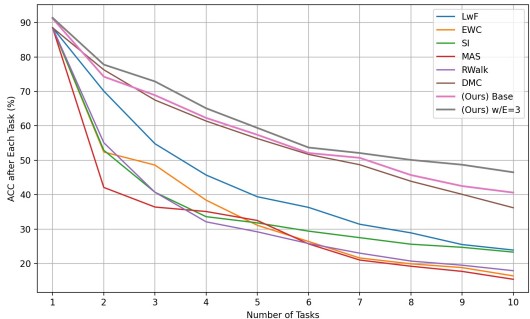

Figure B: Class incremental learning results on *10-Split CIFAR100*.

### C.7.2    ADDITIONAL RESULTS ON *10-Split ImageNet-Subset*

We further validate our method on a large-scale image classification dataset, ImageNet-Subset, which is a subset of the ImageNet dataset that contains the first 100 classes. As in Masanaet et al. (2020), we split 100 classes into 10 tasks with 10 classes per task. We adopt the ResNet-18 as the architecture and substitute the convolutional layers with our decomposed version. Training details are provided in Appendix C.3.

Table F: Class-incremental results on *10-Split ImageNet-Subset*.

| Method | Memory | Acc. (last) |
|---|---|---|
| iCaRL (Rebuffi et al., 2017) | | 43.6 |
| EEIL (Castro et al., 2018) | 301.4 MB | 36.6 |
| BiC (Wu et al., 2019) | | 45.6 |
| IL2M (Belouadah & Popescu, 2019) | | 38.2 |
| Ours (Base) | **7.3 MB** | 43.9 |
| w/ $E_w = 2$ | **7.7 MB** | **46.3** |

We benchmark our method with existing methods that store 2000 exemplars in the external memory. Their results are faithfully reproduced in Masanaet et al. (2020). As shown in the Tab. F, our method with intra-task ensemble achieves the best result with significantly less memory usage.

### C.7.3    ADDITIONAL RESULTS ON CUBS AND FLOWERS

We further validate our methods on CUBS (Welinder et al., 2010) and Flowers (Nilsback & Zisserman, 2008) datasets. Following Yu et al. (2020), we use the random seed (1993) to select 100/50 classes as the first task, and

evenly split the rest 100/50 classes into 5 tasks. We adopt ResNet-18 pretrained on ImageNet with substituted decomposed convolutional layers. Training details are illustrated in Appendix C.3.

Table G: Average incremental accuracy on CUBS and Flowers datsets

| Method | CUBS | Flowers |
|---|---|---|
| LwF-E* (Li & Hoiem, 2017) | 69.8 | 87.2 |
| EWC-E* (Kirkpatrick et al., 2017a) | 69.7 | 85.9 |
| MAS-E* (Aljundi et al., 2018b) | 68.5 | 84.7 |
| SDC (Yu et al., 2020) | 70.0 | 86.8 |
| Ours (Base) | 69.3 | 86.9 |
| w/ $E_w = 2$ | **72.1** | **89.3** |

Under the class-incremental setting, we compare the average incremental accuracy with other benchmarks. On both datasets, as shown in the table above, our method produces the best results.

## C.8 ABLATION STUDY ON DISTANCE METRICS IN INTER-TASK ENSEMBLE

We present an ablation study of different distance metrics including the Grassmann distance, L2 norm, and cosine distance for inter-task ensemble. As shown in Tab. H, inter-task ensemble with Grassmann distance achieves the best result.

Table H: Ablation study on different metrics in inter-task ensemble.

| Method | CIFAR-100 | miniImageNet |
|---|---|---|
| Ours (Base) | 79.13 | 66.01 |
| w/ $E_c = 3$ (L2 norm) | 80.35 | 67.13 |
| w/ $E_c = 3$ (cosine) | 80.11 | 66.82 |
| w/ $E_c = 3$ (**Grassmann**) | **80.75** | **67.84** |

## C.9 INFERENCE TIME COMPARISONS

On the CIFAR-100 dataset, we compare inference time on all classes with other methods. All methods are tested on a single RTX 2080ti GPU under the class-incremental setting. Our method has comparable inference time to other methods, as shown in the Tab. I.

Table I: Inference time comparisons on CIFAR100.

| Method | Time (s) |
|---|---|
| EWC-E* (Kirkpatrick et al., 2017a) | 9.37 |
| LwM-E* (Dhar et al., 2019) | 9.71 |
| iCaRL (Rebuffi et al., 2017) | 10.03 |
| BiC (Wu et al., 2019) | 9.35 |
| Ours (Base) | 10.27 |
| w/ $E_w = 2$ | 13.51 |

## C.10 MEMORY CALCULATION

### C.10.1 CLASS-INCREMENTAL LEARNING

We calculate the external memory usage for exemplar-based methods with 2000 exemplars, e.g., Rebuffi et al. (2017), and our method with atom memory.

- 2000 exemplars (CIFAR-100): $2000 * (32 * 32 * 3 + 1)/1e6 = 6.15$ MB.
- 2000 exemplars (ImageNet-Subset): $2000 * (224 * 224 * 3 + 1)/1e6 = 301.4$ MB.
- **Our (Base) (ResNet-32):** $(31 * 1 * 12 * 9 + 16 * 5 * 9 + 32 * 5 * 9 + 64 * 5 * 9 + 64 * 10 * 9 + 16 * 32 * 9 + 32 * 64 * 9) * 4/1e6 * 9 = 0.19$ **MB.**
- **Our (w/ $E_w = 2$):** $(31 * 1 * 12 * 9 + 16 * 5 * 2 * 9 + 32 * 5 * 2 * 9 + 64 * 5 * 2 * 9 + 64 * 10 * 9 * 2 + 16 * 32 * 9 + 32 * 64 * 9) * 4/1e6 * 9 = 0.52$ **MB.**

### C.10.2 TASK-INCREMENTAL LEARNING

We calculate external memory usage for our method in 20-Split CIFAR100 and 20-Split miniImageNet below.

- **Our (Base):** $(5 * 12 * 9 + 256 * 5)/1e6 * 4 * 19 = 0.14$ MB
- **Our (w/ $E_c = 3$):** $(5 * 12 * 9 + 256 * 5 + 256 * 5 * 3)/1e6 * 4 * 19 = 0.43$ MB
- **Our (w/ $E_w = 2$):** $(5 * 12 * 9 * 2 + 256 * 5 * 2)/1e6 * 4 * 19 = 0.28$ MB
- **Our (w/ both):** $(5 * 12 * 9 * 2 + 256 * 5 * 2 + 256 * 5 * 2 * 3)/1e6 * 4 * 19 = 0.86$ MB

