# OpenReview forum: "Continual Learning with Filter Atom Swapping"
_ICLR.cc/2022/Conference — ICLR 2022 Spotlight_

### Official Review · Reviewer_WerS · 2021-11-01

**Correctness:** 3
**Technical Novelty And Significance:** 3
**Empirical Novelty And Significance:** 3
**Recommendation:** 8
**Confidence:** 4

**Main Review:**

PROs:
- The paper is for most parts well written. With very few exceptions, all the text is clear at a first read. Figures, mathematical notation and equations are properly used and truly help in understanding several technical aspects.
- The method proposed is technically sound. By keeping separate atoms for every task, they do not interfere with each other and therefore do not imply forgetting of past knowledge.
- The results presented for class-incremental learning are very encouraging (Tab. 1), especially considering the model does not involve any buffer, that is considered almost a standard for such settings. Even in its absence, the model can outperform a number of competing methods that store examples from past tasks, at a fraction of the memory overhead.
- To my knowledge, the low-rank decomposition of filters (a standard in other fields such as neural network compression) is novel in the continual learning field. This allows the network to grow as new tasks are encountered, like in other models such as Progressive Neural Networks [1]. However, by restricting the growth to the D components only, memory is impacted in a much more limited way.

---
CONs:
- The class-incremental experiment, the most interesting in my judgement, is only conducted on a single dataset (CIFAR-100). It would be interesting to check whether the same findings hold on miniImageNet as well. Moreover, the authors explain such a setting by stating "does not provide task-id during training". However, the exact definition is that it does not rely on task-id during inference[5,6]. Can the authors confirm this is the case for the results presented in Fig.4?
- A substantial part of Sec. 3 describes several ensembling techniques that are used in different settings within experiments. Although I understand ensembling is beneficial for performances on a single task, I consider this contribution somewhat orthogonal to the main aim of the paper. Indeed, it does not involve any interaction between different tasks during training and it is therefore not related to the sequential nature of the continual learning problem. It seems the authors included that as their model is structurally more prone to ensembling (which I acknowledge it is) that regular CNNs. For this reason, it is a bit difficult to assess this contribution. Couldn't other ensembling techniques be used, both for the proposed model and for competing methods?
- The derived excess risk bound is not commented. The authors should add text describing what the derived bound means and why it is meaningful. Moreover, it is derived under a couple of strong assumptions, namely a single linear layer and a MSE objective, that are not met in the experiments carried out. As such, it is a bit difficult to assess whether that would generalize to deep networks used in practice.
- To my understanding, the model as is does not allow for any forward knowledge transfer, as past Ds are not used during training of the current task. This seems suboptimal, as other "architectural methods" such as [1,2,3,4,5] envision this possibility by allowing past filters to be employed during future tasks. Here, this possibility is only granted by the ensembling mechanism, which is however not explicitly trained for the purpose of forward transfer.
- The authors employ the Grassman distance in ensembling, in order to retrieve past task parameters that are somewhat similar to the parameters of the current task. This choice is not validated in experiments, and other types of distances should be tested, such as L2 or cosine similarity in parameters space.
- In authors' proposal, the matrix alpha is trained on the first task only. This raises the question whether the overall model is susceptible to the choice of the first task. A good way to check this would be to have a set of first tasks $F=\{T_1,...,T_m\}$ and a set of remaining tasks $R=\{T_{m+1}, ..., T_n\}$. If we repeat an experiment where the first task is sampled from F, and then all tasks from R are trained in a sequence, would the outcomes (i.e. performances on $R$, excluding the first warmup task) have a high variance?

Minor, questions:
- After Eq. 5, a $()_+$ sign is explained. However, that does not appear in the equation.
- Referring to Tab. 1, are all regularization-based methods used in a task-incremental way, by excluding classes belonging to unrelated tasks from the prediction?
- In Tab.1, why does HAT have (slight) backward transfer? To my knowledge, weights for the previous tasks are fixed and therefore it should have exactly 0 forgetting and BWT (same as PNN and the model proposed by authors.)

---
References:
- [1] Rusu, Andrei A., et al. "Progressive neural networks." arXiv preprint arXiv:1606.04671 (2016).
- [2] Serra, Joan, et al. "Overcoming catastrophic forgetting with hard attention to the task." International Conference on Machine Learning. PMLR, 2018.
- [3] Mallya, Arun, and Svetlana Lazebnik. "Packnet: Adding multiple tasks to a single network by iterative pruning." Proceedings of the IEEE conference on Computer Vision and Pattern Recognition. 2018.
- [4] Mallya, Arun, Dillon Davis, and Svetlana Lazebnik. "Piggyback: Adapting a single network to multiple tasks by learning to mask weights." Proceedings of the European Conference on Computer Vision (ECCV). 2018.
- [5] Abati, Davide, et al. "Conditional channel gated networks for task-aware continual learning." Proceedings of the IEEE/CVF Conference on Computer Vision and Pattern Recognition. 2020.
- [6] Van de Ven, Gido M., and Andreas S. Tolias. "Three scenarios for continual learning." arXiv preprint arXiv:1904.07734 (2019).

**Summary Of The Paper:**

This paper introduces a model for continual learning based on the decomposition of linear filters into low-rank components, called atoms. Specifically, the authors decompose convolutional filters shaped (c,c',k,k) into two components: i) alpha, shaped (c,c',m) and D, shaped (m,k,k). The former is learned on the first task and then frozen, whereas for D every task has its own and they are do not conflict during optimization. On top of that, the authors envision two different ensembling schemes that improve performances. i) First, in task-incremental settings, they retrieve atoms from the task of interest and from similar tasks as well (based on SVD decomposition of D matrices and Grassman distance) and ensemble them. ii) Furthermore, in class-incremental settings, they explicitly setup multiple atoms per task, building a task ensemble. During inference, all ensembles are queried and their predictive variance is used to "discover" the relevant task, for which a prediction is carried out. Experiments are carried out on 3 datasets in both task-incremental and class-incremental learning settings.


**Summary Of The Review:**

In my opinion, the paper introduces a very simple model for continual learning and showcases extremely encouraging results in class-incremental learning. The submission could further improve by including class-incremental experiments on a second dataset (miniImagenet) and clarifying the points raised above about model ensembling and risk bound.

---- POST REBUTTAL COMMENT ----
As detailed in my answer below, I will increase my score to 8 after reading the author's feedback.

---

> ### Author Response · Authors · 2021-11-16
> **Thank you for your insightful feedback!**
>
> Thank you for your constructive comments. We hope the responses below will alleviate your concerns.
>
>
> **1. Class-incremental experiments**
>
> Thanks for pointing out the typo, it should be 'does not provide task-id during testing'. It has been corrected in the uploaded revision.
>
> Following [1], we further illustrate the effectiveness of our method under the class-incremental setting on the ImageNet-Subset dataset, which is a subset of the ImageNet dataset that contains 100 classes. As in [1], we split 100 classes into 10 tasks with 10 classes per task. We adopt the ResNet-18 as the architecture and substitute the convolutional layers with our decomposed version. Training details are provided in Appendix C.3.
>
>
> |Method | Memory | Acc.(last)|
> |:-----:|:------:|:---------:|
> |EWC-E* [2]     |301.4MB|30.1|
> |LwM-E* [3]     |301.4MB|33.8|
> |iCaRL [4]    |301.4MB|43.6|
> |EEIL [5]       |301.4MB|36.6|
> |BiC [6]        |301.4MB|45.6|
> |IL2M [7]       |301.4MB|38.2|
> |Ours (Base)   |**0.18MB**|43.9|
> |**w/ $E_w=2$**|**0.37MB**|**46.3**|
>
> We benchmark our method with existing methods that store 2000 exemplars in the external memory. Their results are faithfully reproduced in [1]. As shown in the table above, our method with intra-task ensemble achieves the best result with significantly less memory usage.
>
> **2. Ensemble strategies**
>
> Thank you for bringing up the topic of our ensemble strategies. We would like to clarify both ensemble strategies are enabled mostly by our atom-swapping framework. For inter-task ensemble, our framework provides an efficient way to select appropriate ensemble members from the past using filter atom subspace distance. The sequential nature of continual learning facilitates this strategy, as the potential ensemble member pool contains previous models and grows larger with more tasks in the sequence. Other continual learning methods such as replay-based and regularization-based ones do not faithfully preserve models of past tasks, therefore inter-task ensemble is not applicable. Expansion-based methods preserve historical models, but they lack an efficient method for selecting appropriate ensemble members. For intra-task ensemble, the proposed framework enables a highly efficient way to add an ensemble member by adding another set of filter atoms per layer (a few hundred of parameters typically). Other ensemble methods may also be adopted, but will not be as efficient as the proposed one in terms of the number of parameters and inference time.
>
> **3. Excess risk bound and real practice**
>
> As we can see from the excess risk bound, having more data or having less data diversity reduces the empirical risk of our model compared to the optimal one, implying better model performance. It sheds light on the reason why our model maintains accuracy on subsequent tasks with shared coefficients $\alpha$. Also, we simplify the complex analysis of deep networks by reducing it to a tractable one. It better helps us understand how to study the risk bound of an atom-based continual framework.
>
>
> **4. Forward knowledge transfer**
>
> Thanks for bringing out the forward transfer for discussion. In both class-incremental and task-incremental settings, the most recent atoms are used to initialize the atoms of the current task. Past knowledge contained in the atoms is thus transferred to the future.
>
> **5. Choices of distance metric**
>
> We provide the ablation study on the metric selection below. We compare ensemble results with members selected by different distance metrics including L2 norm distance, cosine distance, and the proposed Grassmann distance.
>
> |Method|CIFAR-100|miniImageNet|
> |:----:|:--:|:---:|
> |Ours (Base)|79.13|66.01|
> |w/ $E_c=3$ (L2 norm)|80.35|67.13|
> |w/ $E_c=3$ (cosine)|80.11|66.82|
> |w/ $E_c=3$ (Grassmann)|**80.75**|**67.84**|
>
> As shown in the table above, the proposed atom subspace distance achieves the best performance.
>
>
> **6. Sensitivity to the first task**
>
> The continual learning literature mostly sample tasks from a single dataset [4,5,6]. In this commonly adopted scenario, we observe that our model is not sensitive to the selection of the first task. we provide an ablation study on 20-split CIFAR-100. As suggested, we perform experiments with 20 different selections of the first task and calculate the statistics of the results of 20 runs. Our base method achieves $ACC=79.41\pm0.92$, which shows a small variance.
>
>
>
> **7. Minor questions**
>
> a. We would like to clarify that it's in the equation, $(\alpha \times D^t(x^t))_\mathbf{+}$.
>
> b. Yes, they are used in a task-incremental way.
>
> c. Thank you, and it has been corrected in the revision.

---

> > ### Author Response · Authors · 2021-11-16
> > **References for the rebuttal**
> >
> > #### References
> > 1. Masana, Marc, et al. "Class-incremental learning: survey and performance evaluation on image classification." arXiv preprint arXiv:2010.15277 (2020).
> > 2. Kirkpatrick, James, et al. "Overcoming catastrophic forgetting in neural networks." Proceedings of the national academy of sciences 114.13 (2017): 3521-3526.
> > 3. Dhar, Prithviraj, et al. "Learning without memorizing." Proceedings of the IEEE/CVF Conference on Computer Vision and Pattern Recognition. 2019.
> > 4. Rebuffi, Sylvestre-Alvise, et al. "icarl: Incremental classifier and representation learning." Proceedings of the IEEE conference on Computer Vision and Pattern Recognition. 2017.
> > 5. Castro, Francisco M., et al. "End-to-end incremental learning." Proceedings of the European conference on computer vision (ECCV). 2018.
> > 6. Wu, Yue, et al. "Large scale incremental learning." Proceedings of the IEEE/CVF Conference on Computer Vision and Pattern Recognition. 2019.
> > 7. Belouadah, Eden, and Adrian Popescu. "Il2m: Class incremental learning with dual memory." Proceedings of the IEEE/CVF International Conference on Computer Vision. 2019.

---

> > ### Comment · Reviewer_WerS · 2021-11-30
> > **Post rebuttal comment**
> >
> > I thank the authors for their response and I acknowledge that I am satisfied by most of their answers. Specificaly:
> >
> > - Class incremental experiments on ImageNet-Subset confirm the outstanding results obtained for CIFAR-100. Moreover, within the answer to reviewer B5kZ, they add two more datasets, which is remarkable.
> > - The authors validate the use of the Grassman distance against baseline metrics.
> > - Interestingly, authors show their model is not impacted by the choice of the first task, which is insightful.
> >
> > I retain my concerns about ensembling (that I consider unrelated to continual learning) and about the strong assumptions involved in the empirical risk derivation. However, at this point, such concerns are clearly outnumbered by the strenghts of the paper.
> >
> > As such, I will increase my score.

---

### Official Review · Reviewer_GWGw · 2021-11-01

**Correctness:** 4
**Technical Novelty And Significance:** 3
**Empirical Novelty And Significance:** 4
**Recommendation:** 8
**Confidence:** 3

**Main Review:**

Strengths:
1. This paper is well-written and easy to follow.
2. Learning new tasks on low-rank filter subspace seems interesting and novel for solving continual learning problems.
3. The proposed method achieves good performance on the benchmark dataset.

Weaknesses:
1. For the experimental results, we can see that the SOTA performance of the proposed method is based on the ensemble scheme. However, it may not be fair when adopting the model ensemble strategy when compared with other methods, especially the intra-task ensemble. The intra-task ensemble will linearly increase the memory and computation consumption with the E_w.

2. The proposed low-rank filter strategy would allow highly efficient model storage while it seems that it does not reduce the computation during testing. The authors may need to compare the inference time among different methods.

3. The proposed method is based on the over-parameterization of deep models. If we use a lightweight network architecture, the advantage of the proposed method compared with other expansion-based methods would narrow down.




**Summary Of The Paper:**

This paper tackles the continual learning via enforcing a low-rank filter structure to each CNN layer.  They first perform atom-coefficient filter decomposition and then learn each task with a new filter subspace, so that the method only needs to save the new filters for each task. The contribution of this paper includes the low-rank filter scheme and the designed intra-task and inter-task model ensemble performing on the filters. The proposed method also achieves SOTA performance on several datasets with tiny size of model memory.


**Summary Of The Review:**

This is an interesting paper by introducing the idea of subspace modeling of tasks into continual learning and further proposing two model ensemble strategies to improve the performance. The proposed method also achieve good performance on benchmark datasets.

---

> ### Author Response · Authors · 2021-11-16
> **Thank you for your supportive review!**
>
> Thank you for your supportive comments. We hope the responses below will alleviate your concerns.
>
>
> **1. Intra-task ensemble**
>
> Thanks for suggesting the discussion about the intra-task ensemble. We agree that simply adopting a typical ensemble strategy with a large number of members can be unfair due to a significant increase in memory and computation consumption. However, our intra-task ensemble strategy is only enabled by the proposed atom-swapping continual learning framework, where a new member can be added highly efficiently as each group of filter atoms only contains a few hundred parameters. Moreover, as shown in the experiments, we obtain satisfactory results with two member ensemble ($E_w=2$), which requires only a small amount of extra memory and computation.
>
> **2. Efficiency in terms of inference time**
>
> On the CIFAR-100 dataset, we compare inference time on all classes with other methods. All methods are tested on a single RTX 2080ti GPU under the class-incremental setting. Our method has comparable inference time to other methods, as shown in the table below.
>
> |     Method    |  Time (s)  |
> |:-------------:|:------:|
> | EWC-E* [1] | 9.37 |
> | LwM-E* [2] | 9.71 |
> | iCaRL [3] | 10.03 |
> | BiC [4] | 9.35 |
> | Our (Base)   | 10.27 |
> | w/ $E_w=2$ | 13.51 |
>
>
> **3. Performance with lightweight networks**
>
> Thanks for bringing the lightweight network into discussion. We will certainly explore it more in our future work.
>
>
> #### References:
> 1. Kirkpatrick, James, et al. "Overcoming catastrophic forgetting in neural networks." Proceedings of the national academy of sciences 114.13 (2017): 3521-3526.
> 2. Dhar, Prithviraj, et al. "Learning without memorizing." Proceedings of the IEEE/CVF Conference on Computer Vision and Pattern Recognition. 2019.
> 3. Rebuffi, Sylvestre-Alvise, et al. "icarl: Incremental classifier and representation learning." Proceedings of the IEEE conference on Computer Vision and Pattern Recognition. 2017.
> 4. Wu, Yue, et al. "Large scale incremental learning." Proceedings of the IEEE/CVF Conference on Computer Vision and Pattern Recognition. 2019.

---

### Official Review · Reviewer_B5kZ · 2021-11-04

**Correctness:** 3
**Technical Novelty And Significance:** 3
**Empirical Novelty And Significance:** 2
**Recommendation:** 6
**Confidence:** 3

**Main Review:**

In general, the idea of the article is concise and clear. The survey of related work and experiment settings are sufficiently detailed. More importantly, there is a huge improvement in memory saving compared with other methods.

However, there are two main problems that I am concerned about:
i. In Sec 5.2.2, there are some recent methods that show better performance on Cifar100(20 tasks) in Paper with code. But the author doesn't consider them, such as "Understanding Catastrophic Forgetting and Remembering in Continual Learning with Optimal Relevance Mapping", "Compacting, Picking and Growing for Unforgetting Continual Learning", etc.
ii. I think only the results of the two datasets are not enough to support the author's conclusion strongly. The author would better add a few more experiments, and there are plenty of datasets shown in other papers for your choices, such as CUBS, Stanford Cars and Flowers, etc.

**Summary Of The Paper:**

The paper, motivated by the task subspace modeling literature, enforced a low-rank filter structure to each CNN layer across time in continual learning. It not only ensures that the knowledge of the past tasks is not lost but also saves a lot of computing memory. Meanwhile, the paper proposes novel intra-task ensembles and inter-task ensembles for class-incremental settings and task-incremental settings, respectively.

**Summary Of The Review:**

The paper idea is creative and has made a significant improvement in saving memory, but more experiments are still needed to support the conclusion.

---

> ### Author Response · Authors · 2021-11-16
> **Thank you for your thorough review!**
>
> Thank you for your constructive comments. We hope the responses below will alleviate your concerns.
>
>
> **1. Other benchmarks on CIFAR100 results**
>
> Thanks for providing these related works. We would like to point out that these works do not serve as a fair comparison due to significantly larger networks are adopted there. More specifically, ResNet-18 and VGG16-BN are used respectively, which are 18 and 16-layer networks, whereas our method adopts a AlexNet-like network with only 6 layers.
>
> **2. Results on additional datasets**
>
> We have additional results on 5-Split MNIST in Appendix Tab. D, where our base method outperforms other methods.
>
> As requested, we provide below additional experiments on CUBS-200-2011 [1] and Flowers-102 [2] datasets. Following [3], we use the random seed (1993) to select 100/50 classes as the first task and evenly split the rest 100/50 classes into 5 tasks. Training details are illustrated in Appendix C.3.
>
> |Method|CUBS|Flowers|
> |:----:|:--:|:-----:|
> |LwF-E* [4]| 69.8 | 87.2 |
> |EWC-E* [5]|69.7|85.9|
> |MAS-E* [6]|68.5|84.7|
> |SDC [3]|70.0|86.8|
> |**Ours (Base)**|69.3|86.9|
> |**w/ $E_w=2$** |**72.1**|**89.3**|
>
>
> Under the class-incremental setting, we compare the average incremental accuracy with other benchmarks. On both datasets, as shown in the table above, our method produces the best results.
>
>
> #### References:
> 1. Wah, Catherine, et al. "The caltech-ucsd birds-200-2011 dataset." (2011).
> 2. Nilsback, Maria-Elena, and Andrew Zisserman. "Automated flower classification over a large number of classes." 2008 Sixth Indian Conference on Computer Vision, Graphics & Image Processing. IEEE, 2008.
> 3. Yu, Lu, et al. "Semantic drift compensation for class-incremental learning." Proceedings of the IEEE/CVF Conference on Computer Vision and Pattern Recognition. 2020.
> 4. Li, Zhizhong, and Derek Hoiem. "Learning without forgetting." IEEE transactions on pattern analysis and machine intelligence 40.12 (2017): 2935-2947.
> 5. Kirkpatrick, James, et al. "Overcoming catastrophic forgetting in neural networks." Proceedings of the national academy of sciences 114.13 (2017): 3521-3526.
> 6. Aljundi, Rahaf, et al. "Memory aware synapses: Learning what (not) to forget." Proceedings of the European Conference on Computer Vision (ECCV). 2018.

---

### Official Review · Reviewer_ymsc · 2021-11-08

**Correctness:** 3
**Technical Novelty And Significance:** 3
**Empirical Novelty And Significance:** 4
**Recommendation:** 8
**Confidence:** 4

**Main Review:**

(+) Interesting idea of considering filter subspace with the construction of filter atoms and sharing the subspace coefficient fixed while changing the small set of filter atoms corresponding to the task.
(-) Different distance metrics other than Grassmann should be considered to see whether it leads to consistent performance. the computational required to compute the distance between filter subspaces requires calculating an SVD which is computationally heavy.
(-) The performance of the proposed approach will depend on the dataset at hand. For very diverse tasks, an inter-task ensemble would not be very effective.
(-)The performance of the proposed algorithm can vary heavily on the diversity of tasks.
(-) (Introduction) I do not agree with the first sentence of the introduction. Humans do forget concepts learned in the past.



**Summary Of The Paper:**

The paper proposes a continual learning algorithm that enforces the convolutional filter in each layer to a low-rank filter subspace defined by a small set of filter atoms. For each task, each convolutional layer is defined by a new filter subspace but subspace coefficients are shared among the tasks. The algorithm is validated on multiple benchmark datasets.

**Summary Of The Review:**

The paper is written and organized adequately. The paper is based on extensive experimental results. The idea is relatively simple and performs well.

---

> ### Author Response · Authors · 2021-11-16
> **Thank you for your supportive review!**
>
> Thank you for your supportive comments. We hope the responses below will alleviate your concerns.
>
> **1. Different metrics for inter-task ensemble**
>
> We adopt the Grassmann distance, which measures the filter atom subspace distance by finding pairs of minimum-angle principal vectors. We present an ablation study of distance metrics below and observe superior performance from the Grassmann distance than L2 norm and cosine distance.
>
> |Method|CIFAR-100|miniImageNet|
> |:----:|:-------:|:----------:|
> |Ours (Base)|79.13|66.01|
> |w/ $E_c=3$ (L2 norm)|80.35|67.13|
> |w/ $E_c=3$ (cosine)|80.11|66.82|
> |w/ $E_c=3$ (Grassmann)|**80.75**|**67.84**|
>
> Moreover, the SVD is only computed for matrices of a size less than $9\times9$, which only introduces negligible computation cost.
>
>
> **2. Performance on very diverse tasks**
>
> The continual learning literature mostly sample tasks from a single dataset [1,2,3]. In this commonly adopted scenario, we observe inter-task ensemble leads to consistent performance gain via retrieving models with similar predictive abilities from the past. When the diversity of tasks becomes sufficiently large, one may consider learning multiple coefficients $\alpha$, each of which can handle a subset of tasks with acceptable diversity. Thanks for raising this outstanding point, and we will for sure explore it in our future work.
>
>
> **3. The first sentence in the Introduction**
>
> It has been revised in the updated version.
>
>
> #### References:
> 1. Rebuffi, Sylvestre-Alvise, et al. "icarl: Incremental classifier and representation learning." Proceedings of the IEEE conference on Computer Vision and Pattern Recognition. 2017.
> 2. Wu, Yue, et al. "Large scale incremental learning." Proceedings of the IEEE/CVF Conference on Computer Vision and Pattern Recognition. 2019.
> 3. Prabhu, Ameya, Philip HS Torr, and Puneet K. Dokania. "Gdumb: A simple approach that questions our progress in continual learning." European conference on computer vision. Springer, Cham, 2020.

---

### Decision · Program_Chairs · 2022-01-20

**Decision:**

Accept (Spotlight)

**Comment:**

The authors propose a memory-based continual learning method that decomposes the models' parameters and that shares a large number of the decomposed parameters across tasks. In other words, only a small number of parameters are task-specific and the memory usage of storing models from previous tasks is hence a fraction of the memory usage of previous approaches. The authors take advantage of their method to propose specific ensembling approaches and demonstrate the strong performance of their methods using several datasets.

In the rebuttal, the authors were very reactive and provided many useful additional results during including a comparison of the computational cost of their method vs. others, results using two new datasets (CUBS & Flowers), and additional results on mini-ImageNet. They also answered, through additional experiments, several reviewer questions including the robustness to different first tasks in the sequence.

Overall, the reviewers after the rebuttal/discussion period agree that this is a strong contribution: novel and fairly simple method with some theoretical justification, thorough empirical evaluation, well-written and easy to follow manuscript. It also opens a few interesting avenues some of which the authors have already explored in their paper (e.g., ensembling).